# Should the Splenic Vein Be Preserved—Fate of Sinistral Portal Hypertension after Pancreatoduodenectomy with Vascular Re-Section for Pancreatic Cancer

**DOI:** 10.3390/cancers14194853

**Published:** 2022-10-04

**Authors:** Sung Hyun Kim, Seung-Seob Kim, Ho Kyoung Hwang, Woo Jung Lee, Chang Moo Kang

**Affiliations:** 1Department of Hepatobiliary and Pancreatic Surgery, Yonsei University College of Medicine, Seoul 03722, Korea; 2Pancreaticobiliary Cancer Center, Yonsei Cancer Center, Severance Hospital, Seoul 03722, Korea; 3Department of Radiology and Research Institute of Radiological Science, Severance Hospital, Yonsei University College of Medicine, Seoul 03722, Korea

**Keywords:** pancreatic cancer, sinistral portal hypertension, splenic vein ligation, survival, varices

## Abstract

**Simple Summary:**

This study aims to evaluate sinistral portal hypertension (SPH) development and its clinical impact on the long-term outcomes of patients with pancreatic cancer who underwent surgical resection with splenic vein (SV) ligation. Data from 94 consecutive patients who underwent pancreatoduodenectomy (PD) with vascular resection for pancreatic cancer were divided into two groups according to SV ligation, and the groups were compared. Variceal score in the SV ligation group was significantly higher than that in the SV saving group at the same postoperative periods, and clinically relevant variceal bleeding was noted in one patient from the SV ligation group. In survival analysis, there was no significant difference between the two groups. These results showed that although SV ligation induced SPH during PD for pancreatic cancer, it did not lead to clinically significant long-term complications. In addition, it did not impact the long-term survival of patients with resected pancreatic head cancer.

**Abstract:**

Background: This study aims to evaluate sinistral portal hypertension (SPH) development and its clinical impact on the long-term outcomes of patients with pancreatic cancer who underwent surgical resection with splenic vein (SV) ligation. Methods: Data from 94 consecutive patients who underwent pancreatoduodenectomy (PD) with vascular resection for pancreatic cancer from 2008 to 2019 were retrospectively collected. The patients were divided into two groups according to whether the SV was preserved or ligated during the surgery. Their computed tomography images were serially reviewed (preoperative, 6-, 12-, and 24-months postoperative) with clinical parameters. The degree of variceal formation (variceal score) and splenomegaly were assessed, and the oncologic outcomes were compared between the two groups. Variceal score in the SV ligation group was significantly higher than that in the SV saving group at the same postoperative periods (SV saving vs. ligation: 12 months, 0.9 ± 1.3 vs. 3.5 ± 2.2, *p* < 0.001; 24 months, 1.4 ± 1.8 vs. 4.0 ± 3.4, *p* = 0.009). Clinically relevant variceal bleeding was noted in one patient from the SV ligation group (SV saving vs. ligation: 0.0% vs. 3.1%, *p* = 0.953). In survival analysis, there was no significant difference between the two groups (DFS; SV saving vs. ligation: 13.0 (11.1–14.9) months vs. 13.0 (10.4–15.6) months, *p* = 0.969, OS; SV saving vs. ligation: 35.0 (19.9–50.1) months vs. 27.0 (11.6–42.4) months, *p* = 0.417). Although SV ligation induced SPH during PD for pancreatic cancer, it did not lead to clinically significant long-term complications. In addition, it did not impact the long-term survival of patients with resected pancreatic head cancer.

## 1. Introduction

Pancreatic cancer is one of the most fatal diseases. Over the decades, diagnostic approaches, perioperative management, and adjuvant therapies have advanced. However, the 5-year survival rate of pancreatic cancer is still about 10% [1]. Although surgical resection remains the only treatment with curative potential, only 20% of patients can undergo surgery at the time of diagnosis [2].

Fortunately, neoadjuvant chemotherapy for localized pancreatic cancer patients has improved the surgical resection rate by downgrading the tumor stage [3]. Several studies have shown that neoadjuvant chemotherapy increased the resection rate for borderline resectable pancreatic cancer to greater than 50% [4]. Neoadjuvant chemotherapy, followed by surgical resection, has been adopted as a standard strategy for borderline resectable pancreatic cancer in the National Comprehensive Cancer Network (NCCN) guidelines [5].

As more patients undergo surgical resection, the rate of combined vascular resection has also increased. The superior mesenteric vein (SMV), or portal vein (PV), is the most frequently involved structure, due to its close anatomical location with the pancreas, and the splenic vein (SV) is also sometimes ligated during vascular resection to achieve a margin-negative resection [6].

During vascular resection, SV ligation can induce sinistral portal hypertension (SPH), which is otherwise referred to as left-sided portal hypertension [7]. SPH may cause variceal formation, which can potentially lead to variceal rupture [8]. Therefore, some surgeons suggest several surgical techniques, such as interposition grafting or bypass, during vascular resection to prevent variceal formation [9].

However, only a few studies have evaluated the long-term follow-up of SPH after splenic vein ligation and its clinical significance in oncologic outcomes [10]. We hypothesized that the impact of SPH on the survival of patients would not be significant, considering the notoriously poor prognosis of pancreatic cancer. This study aims to evaluate SPH development and its clinical impact on the long-term outcomes in pancreatic cancer patients who underwent surgical resection with SV ligation. 

## 2. Materials and Methods

### 2.1. Enrolled Patients

Consecutive patients at a single tertiary center who underwent pylorus-preserving pancreatoduodenectomy (PD) with vascular resection (PV or SMV) for localized pancreatic cancer, from January 2008 to December 2019, were retrospectively collected. Among them, we excluded patients who had no available preoperative and six-month postoperative computed tomography (CT) images. Before the exclusion, the medical records were reviewed to determine whether or not variceal bleeding had occurred within 6 months after the surgery. In addition, we excluded patients who showed variceal formation on the preoperative CT images. In total, 94 patients were enrolled in the study.

### 2.2. Clinicopathological Data Collection

We retrieved clinicopathological data, including neoadjuvant or adjuvant chemotherapy, and data on variceal bleeding. We also reviewed the surgery notes and pathologic reports to record the vascular resection method, whether the SV was preserved or ligated, the achievement of R0 resection, and the TNM stage according to the American Joint Committee on Cancer 8th edition [11]. Complications were defined using the Clavien–Dindo classification, and postoperative pancreatic fistula was defined using the classifications of the International Study Group of Pancreatic Fistula [12,13]. Disease-free survival (DFS) and overall survival (OS) were also calculated.

### 2.3. Imaging Assessment

Our radiologist (Kim SS, with seven years of experience in abdominal imaging) consecutively reviewed the preoperative and postoperative CT images. Inferior mesenteric vein (IMV) insertion types were defined as follows: type I, when the IMV drains into the PV/SMV junction; type II, when the IMV drains into the SMV; and type III, when the IMV drains into the SV (Appendix A) [14]. Postoperative ascites and vascular thrombosis were determined based on the immediate postoperative CT images, while the stricture of PV or SMV was evaluated in the six-month postoperative CT images. 

Variceal formation and splenomegaly were assessed in the preoperative and six-month postoperative CT images. The variceal grade was defined according to the number of dilated vessels on the axial planes, and measured as the largest diameter of each of the following portosystemic varices: esophageal, paraesophageal, gastric submucosal, gastric adventitial, splenic, mesenteric, and retroperitoneal varices (Appendix A) [15]. For esophageal, paraesophageal, and gastric submucosal varices, the variceal score was determined from the largest variceal diameter as follows: 0, shorter than 2 mm; 1, 2–2.9 mm; 2, 3–6.9 mm; and 3, longer than 7 mm. For the gastric adventitial, splenic, mesenteric, and retroperitoneal varices, the variceal score was determined from the largest variceal diameter as follows: 0, 3 mm; 1, 3–4.9 mm; 2, 5–9.9 mm; and 3, longer than 10 mm. When the number of dilated vessels was more than 4, the variceal score was increased by 1 [15]. The sum of all seven variceal scores was used for statistical analysis. CT images from representative cases with variceal scoring are shown in the Appendix A. A freehand region-of-interest was drawn for the spleen on each axial slice, and the total splenic volume was automatically calculated using commercially available software (Aquarius iNtuition, version 4.4.13.P6, TeraRecon, San Mateo, CA, USA). Variceal scoring and splenic volume measurements were also performed on the 12- and 24-month postoperative CT images, where available.

### 2.4. Prediction Model for Postoperative Variceal Formation

In this study, SPH was defined as the occurrence of left-sided variceal formation, and the variceal formation was regarded as present when the sum of variceal scores was higher than 4 in the postoperative six-month CT images. We evaluated whether the IMV insertion type, postoperative PV or SMV stricture, the vascular resection method, or venous ligation was associated with postoperative variceal formation. 

### 2.5. Comparison of Varix-Related Parameters and Oncologic Outcomes According to SV Status

To evaluate the effect of SV ligation without confounding factors, the subgroup of patients who did not show PV or SMV stricture in postoperative six-month CT images was identified. They were divided into two groups according to SV preservation or ligation during surgery. The varix-related parameters (platelet count ratio, splenic volume ratio, and sum of variceal scores) were compared between the two groups based on preoperative and postoperative 6-, 12-, and 24-month CT images. The platelet count ratio and splenic volume ratio were calculated by division with preoperative values. The number of variceal bleeding events and the oncologic outcomes by DFS and OS were also compared between the two groups. 

### 2.6. Statistics

R version 4.2.1 was used for all statistical analyses. Nominal data were compared using the *χ*^2^ (chi-square) test, while continuous parametric data were compared using the *t*-test. In addition, non-parametric data were compared using the Mann–Whitney test. Logistic regression was used to evaluate prognostic parameters related to variceal formation. Survival parameters were assessed by the Kaplan–Meier method and compared using the log-rank test. A multivariate Cox proportional hazards model was used to evaluate the potential independent effects of the SV status. Parameters with *p*-values < 0.05 in univariate analysis were included in multivariate analysis. The criterion for statistical significance was a *p*-value < 0.05.

## 3. Results

### 3.1. Clinicopathological Characteristics

The clinicopathological characteristics of the enrolled 94 patients are shown in Table 1. Their mean age was 62.2 ± 9.1 years. About half (51.1%, *n* = 48) of the patients were given IMV insertion type III (drain into SV) [14]. In addition, the right hepatic artery branched from the superior mesenteric artery in 14 (14.9%) patients. Luminal stricture of the PV or SMV was present in the six-month postoperative CT images of 26 (27.7%) patients, and 22 (23.4%) patients showed variceal formation. During PD, the vein was resected segmentally in 65 (69.1%) patients, and tangentially in 29 (30.9%) patients. In the tangential resection group, all patients went through transverse suture reconstruction without a patch. In the segmental resection group, 2 (3.1%) patients had vein reconstruction using autologous vein grafts, and the other 63 (96.9%) patients had end-to-end vein anastomosis. The SV was ligated in 50 (53.2%) patients, whereas it was preserved in the other 44 (46.8%) patients during surgery, including in 2 patients who underwent vascular re-anastomosis of the SV and SMV. No patient underwent combined organ resection, such as of the stomach or colon. Seventeen (18.1%) patients underwent R1 resection, and one of these patients showed cancer cells on the vascular resection margin. Although five (5.3%) patients showed vascular thrombosis during the immediate postoperative period, the thrombosis disappeared within six months. There was no 90-day mortality event. Two (2.1%) patients experienced variceal bleeding (both were from the SV ligation group), and the bleeding was controlled using a vigorous diagnostic approach (VDA). The median follow-up duration was 25 (16–37) months.

### 3.2. Prediction Model for Postoperative Variceal Formation

In the univariate analysis, IMV ligation, SV ligation, postoperative PV or SMV stricture, and the vascular resection method (tangential resection vs. segmental resection) were associated with postoperative variceal formation. In the multivariate analysis, however, SV ligation and postoperative PV or SMV stricture were independently associated with variceal formation after surgery (SV ligation: OR = 4.56 (1.53–13.00), *p* = 0.006; and PV or SMV stricture: OR = 3.98 (1.36–11.69), *p* = 0.012) (Table 2).

### 3.3. Characteristics between SV Saving and SV Ligation

Excluding the patients with postoperative PV or SMV stricture (*n* = 26, 15 patients in SV saving group and 11 patients in SV ligation group), the basal characteristics of the remaining 68 patients were compared according to SV status. There was no significant difference between the two groups, except for IMV ligation, the vascular resection method, operation time, and variceal formation (SV saving vs. SV ligation: IMV ligation, 0 (0.0%) vs. 23 (71.9%), *p* < 0.001; vascular resection method (tangential resection: segmental resection), 24:12 vs. 1:31, *p* < 0.001; operation time (min), 453 ± 93 vs. 510 ± 117, *p* = 0.028; and variceal formation, 2 (5.6%) vs. 9 (28.1%), *p* = 0.028) (Table 3).

### 3.4. Comparison of Varix-Related Parameters According to SV Status

The splenic volume ratio of the SV ligation group was significantly higher than that of the SV saving group in 6-, 12-, and 24-month postoperative CT images (SV saving vs. SV ligation: 6 months, 1.02 ± 0.24 vs. 1.42 ± 0.35, *p* < 0.001; 12 months, 1.04 ± 0.34 vs. 1.45 ± 0.34, *p* < 0.001; and 24 months, 1.04 ± 0.44 vs. 1.53 ± 0.43, *p* = 0.003). The variceal score of the SV ligation group was also significantly higher than that of the SV saving group at the same postoperative periods (SV saving vs. SV ligation: 6 months, 0.9 ± 1.2 vs. 3.5 ± 2.2, *p* < 0.001; 12 months, 0.9 ± 1.3 vs. 3.5 ± 2.2, *p* < 0.001; and 24 months, 1.4 ± 1.8 vs. 4.0 ± 3.4, *p* = 0.009). However, the platelet count ratio was not significantly different between the two groups (Table 4, Figure 1).

### 3.5. Oncologic Outcomes According to SV Status and SPH

In the DFS analysis, there was no significant difference between the two groups (SV saving vs. SV ligation: 13.0 (11.1–14.9) months vs. 13.0 (10.4–15.6) months, *p* = 0.969). Similarly, in the OS analysis, there was no significant difference between the two groups (SV saving vs. SV ligation: 35.0 (19.9–50.1) months vs. 27.0 (11.6–42.4) months, *p* = 0.417) (Figure 2). According to the multivariate analyses for DFS and OS, the TNM stage was the only independent poor prognostic factor. SPH did not affect survival after surgery (DFS; HR = 1.21 (0.69–2.14), *p* = 0.509, and OS; HR = 1.03 (0.62–1.73), *p* = 0.902) (Table 5).

## 4. Discussion

SPH can result from thrombosis or obstruction in the SV and lead to back pressure changes in the left portal system [16]. As the cases of pancreatectomy with combined vascular resection increase with improved postsurgical survival, the incidence of SPH is also increasing and has become a concerning issue in pancreatic cancer surgery. However, in our study, although a higher variceal score and splenic volume were observed when the SV was ligated, variceal bleeding and the oncologic outcomes of DFS and OS were not significantly different with respect to the SV status. 

Previous studies have consistently reported that SV ligation during PD was associated with the development of SPH [7,8,9,10,17,18,19,20,21,22,23]. However, the rate of variceal bleeding, the most critical complication of SPH, was variable among the authors (0~10%) [7,8,17,19,20,21,22]. Furthermore, there is no clear consensus regarding SV ligation or preservation during PD. Some authors insist that SV should be either preserved or reconstructed if ligated [21], while others suggest that SV reconstruction is unnecessary [22]. Tanaka et al. suggested that a more tailored approach than SV ligation or reconstruction should be determined based on the number of “critical veins”, such as the superior right colic vein arcade, the middle colic vein, or the left gastric vein [8]. We observed that the SV ligation group had a higher variceal score and splenic volume than the SV preservation group. Nevertheless, among the patients without postoperative PV or SMV stricture, only one (3.1%) experienced variceal bleeding after SV ligation during the PD, the incidence of which was not significantly different from the SV preservation group. More importantly, the oncologic outcomes of DFS and OS did not differ with respect to the SV status in our study. One possible explanation for this result might be the poor survival outcome of pancreatic cancer. According to previous reports, on average, variceal bleeding occurred about two years after the PD [19,24]. However, in our study, more than half of the enrolled patients expired within two years after PD. In this regard, our results suggest that the survival impact of SPH and variceal bleeding are not significant in cases of pancreatic cancer, although variceal bleeding is a well-established poor prognostic factor in other medical fields [25]. 

Fortunately, a few patients showed a good prognosis without recurrence, and their TNM stage was early, such as stage I. Six patients even revealed complete remission (T0), and it seemed that they achieved down-staging following neoadjuvant chemotherapy [26]. Of course, it would have been better if the vascular structures were preserved in these patients. In other words, surgeons did not know the pathologic character at the time of surgery. Because of neoadjuvant chemotherapy, the surroundings of the tumor area showed fibrotic change in the operative field, and it was difficult for the surgeons to discriminate between fibrotic adhesion and tumor invasion of the vascular structure. In that situation, the surgeons had no choice but to perform vascular resection, because they were unsure of whether there was invasion or not. Efforts to discriminate between fibrotic change and tumor invasion are needed for better outcomes.

Whether or not the IMV belongs to “critical veins” is still controversial [7,19,20]. In our study, neither IMV insertion type nor ligation were independently associated with variceal formation after PD. Instead, postoperative stricture of PV or SMV was the independent predictor of variceal formation. Vascular stricture can develop as a complication after vascular anastomosis. It is well known that PV/SMV stricture can cause portal hypertension and eventually lead to graft failure after liver transplantation [27]. However, the clinical significance of postoperative PV/SMV stricture has been rarely reported in studies on pancreatic cancer surgery. Our data suggest that surgeons should make an effort to avoid PV/SMV stricture when performing vascular resection with PV/SMV anastomosis during PD.

While variceal formation and splenomegaly have been consistently observed in patients with SPH, the incidence of thrombocytopenia can vary [17,21,28]. In our study, the platelet count ratio was not significantly different with respect to the SV status, unlike the splenic volume ratio or variceal score. Since thrombocytopenia can be induced by chemotherapy, the association between thrombocytopenia and SPH should be further investigated in a future study [28].

Our study had several limitations. First, the study sample was retrospectively enrolled from a single institution. Therefore, some data depended on subjective records such as operative notes, and there was a selection bias for the surgery decisions. Second, the study sample was relatively small in number. For example, only two cases of variceal bleeding were present, which possibly lowered the statistical power of the study. For this reason, statistically unclear outcomes were obtained in some parameters. Third, over the 10 years of the study period, there have been many changes in surgical indications, equipment, and perioperative management, which may have potentially affected our analysis. A large-scale prospective study is warranted to confirm our conclusions.

## 5. Conclusions

In conclusion, although SV ligation during PD for pancreatic cancer induced SPH, it did not impact the long-term survival of patients with resected pancreatic head cancer. We carefully suggest that surgeons consider SV ligation when it is necessary for R0 resection of pancreatic cancer.

## Figures and Tables

**Figure 1 cancers-14-04853-f001:**
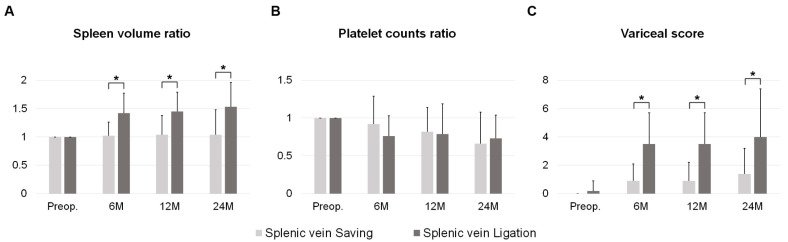
Pre- and postoperative changes in variceal-related parameters: (**A**) spleen volume ratio, (**B**) platelet count ratio, and (**C**) variceal score (* *p* < 0.05).

**Figure 2 cancers-14-04853-f002:**
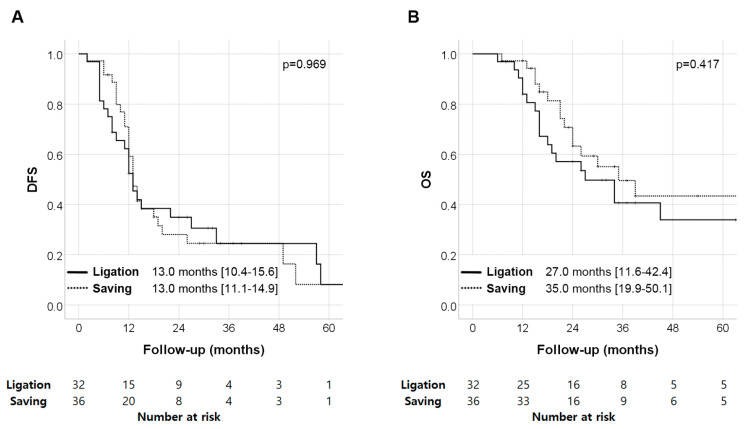
Survival analyses according to splenic vein status: (**A**) disease-free survival and (**B**) overall survival.

**Table 1 cancers-14-04853-t001:** Basal characteristics of the enrolled patients.

	*n* = 94
Age (years)	62.2 ± 9.1
Sex (M:F)	52:42
BMI (kg/m^2^)	22.8 ± 3.1
Liver diseases	6 (6.4%)
CA 19-9 (U/mL)	98 (15–416)
Platelet count (10^3^/uL)	249 ± 87
Splenic volume (cc)	156 ± 101
Neoadjuvant chemotherapy	45 (47.9%)
IMV insertion type	
Type I	8 (8.5%)
Type II	38 (40.4%)
Type III	48 (51.1%)
Postoperative PV or SMV stricture	26 (27.7%)
Vascular resection method	
Tangential resection	29 (30.9%)
Segmental resection	65 (69.1%)
Splenic vein status	
Splenic vein ligation	50 (53.2%)
Splenic vein saving	44 (46.8%)
IMV ligation	34 (36.2%)
Operation time (min)	487 ± 116
Blood loss (mL)	888 ± 613
Complication (CDC ≥ III)	12 (12.8%)
POPF (≥Grade B)	2 (2.1%)
Ascites	12 (12.8%)
Thrombosis	5 (5.3%)
R1 resection	17 (18.1%)
T stage	
0	6 (6.4%)
1	30 (31.9%)
2	54 (57.4%)
3	4 (4.3%)
N stage	
0	43 (45.7%)
1	39 (41.5%)
2	12 (12.8%)
TNM stage	
0	6 (6.4%)
I	36 (38.3%)
II	40 (42.6%)
III	12 (12.8%)
Adjuvant chemotherapy	74 (78.7%)
Variceal formation	22 (23.4%)
Variceal bleeding	2 (2.1%)
Median follow-up (months)	25 (16–37)

BMI, body mass index; CA, carbohydrate antigen; IMV, inferior mesenteric vein; PV, portal vein; SMV, superior mesenteric vein; CDC, Clavien–Dindo classification; POPF, postoperative pancreatic fistula.

**Table 2 cancers-14-04853-t002:** Univariable and multivariable analyses for variceal formation after pancreatoduodenectomy.

		Univariate Analysis			Multivariate Analysis		
		OR	95%CI	*p* Value	OR	95%CI	*p* Value
IMV insertion type	Type I	0.93	0.35–2.51	0.931			
	Type II	0.43	0.05–3.85	0.429			
	Type III	(reference)					
IMV ligation		4.25	1.55–11.63	0.005			
SV ligation		4.29	1.54–11.91	0.005	4.56	1.53–13.00	0.006
Postoperative PV or SMV stricture		3.80	1.38–10.44	0.010	3.98	1.36–11.69	0.012
Vascular resection method	SR	6.00	1.30–27.71	0.022			
	TR	(reference)					

OR, odds ratio; CI, confidence interval; IMV, inferior mesenteric vein; SV, splenic vein; PV, portal vein; SMV, superior mesenteric vein; SR, segmental resection; TR, tangential resection.

**Table 3 cancers-14-04853-t003:** Basal characteristics according to splenic vein status.

	SV Saving (*n* = 36)	SV Ligation (*n* = 32)	*p* Value
Age (years)	62.5 ± 9.2	62.9 ± 9.0	0.866
Sex (M:F)	20:16	14:18	0.466
BMI (kg/m^2^)	23.4 ± 3.2	22.5 ± 2.3	0.191
Liver diseases	5 (13.9%)	1 (3.1%)	0.257
CA 19-9 (U/mL)	73 [10–391]	86 [18–276]	0.907
Neoadjuvant chemotherapy	15 (41.7%)	15 (46.9%)	0.852
IMV insertion type			0.600
Type I	2 (5.6%)	3 (9.4%)	
Type II	17 (47.2%)	11 (34.4%)	
Type III	17 (47.2%)	18 (56.3%)	
IMV ligation	0 (0.0%)	23 (71.9%)	<0.001
Vascular resection type			<0.001
Tangential resection	24 (66.7%)	1 (3.1%)	
Segmental resection	12 (33.3%)	31 (96.9%)	
Operation time (min)	453 ± 93	510 ± 117	0.028
Blood loss (mL)	843 ± 645	978 ± 622	0.385
Complication (CDC≥III)	5 (13.9%)	5 (15.6%)	>0.999
POPF (≥Grade B)	1 (2.8%)	1 (3.1%)	>0.999
Ascites	4 (11.1%)	4 (12.5%)	>0.999
Thrombosis	2 (5.6%)	1 (3.1%)	>0.999
R1 resection	7 (19.4%)	6 (18.8%)	>0.999
T stage			0.967
0	2 (5.6%)	1 (3.1%)	
1	12 (33.3%)	11 (34.4%)	
2	21 (58.3%)	18 (56.3%)	
3	1 (2.8%)	2 (6.3%)	
N stage			0.514
0	14 (38.9%)	13 (40.6%)	
1	15 (41.7%)	16 (50.0%)	
2	7 (19.4%)	3 (9.4%)	
TNM stage			0.655
0	2 (5.6%)	1 (3.1%)	
I	12 (33.3%)	12 (37.5%)	
II	15 (41.7%)	16 (50.0%)	
III	7 (19.4%)	3 (9.4%)	
Adjuvant chemotherapy	31 (86.1%)	25 (78.1%)	0.587
Variceal formation	2 (5.6%)	9 (28.1%)	0.028
Variceal bleeding	0 (0.0%)	1 (3.1%)	0.953

SV, splenic vein; BMI, body mass index; CA, carbohydrate antigen; IMV, inferior mesenteric vein; CDC, Clavien–Dindo classification; POPF, postoperative pancreatic fistula.

**Table 4 cancers-14-04853-t004:** Pre- and postoperative changes in variceal-related parameters.

	Parameters	SV Saving (*n* = 36)	SV Ligation (*n* = 32)	*p* Value
Preop.	Splenic volume (cc)	149 ± 71	150 ± 68	0.938
	Platelet counts (10^3^/uL)	269 ± 96	227 ± 64	0.038
	Platelet count < 100K	0 (0.0%)	0 (0.0%)	1.000
	Variceal score	0.0 ± 0.0	0.2 ± 0.7	0.091
6M	Splenic volume (cc)	149 ± 78	213 ± 108	0.007
	Splenic volume (ratio)	1.02 ± 0.24	1.42 ± 0.35	<0.001
	Platelet count (10^3^/uL)	227 ± 76	166 ± 54	<0.001
	Platelet count (ratio)	0.92 ± 0.37	0.76 ± 0.27	0.059
	Platelet count < 100K	0 (0.0%)	2 (6.3%)	0.422
	Variceal score	0.9 ± 1.2	3.5 ± 2.2	<0.001
12M	Splenic volume (cc)	137 ± 61	218 ± 114	0.002
	Splenic volume (ratio)	1.04 ± 0.34	1.45 ± 0.34	<0.001
	Platelet count (10^3^/uL)	205 ± 56	161 ± 56	0.005
	Platelet count < 100K	1 (3.1%)	5 (18.5%)	0.129
	Platelet count (ratio)	0.82 ± 0.32	0.79 ± 0.40	0.677
	Variceal score	0.9 ± 1.3	3.5 ± 2.2	<0.001
24M	Splenic volume (cc)	154 ± 160	250 ± 138	0.074
	Splenic volume (ratio)	1.04 ± 0.44	1.53 ± 0.43	0.003
	Platelet count (10^3^/uL)	184 ± 41	154 ± 54	0.086
	Platelet count < 100K	0 (0.0.%)	1 (6.3%)	0.975
	Platelet count (ratio)	0.66 ± 0.42	0.73 ± 0.31	0.583
	Variceal score	1.4 ± 1.8	4.0 ± 3.4	0.009

**Table 5 cancers-14-04853-t005:** Univariate and multivariate risk of overall survival.

			Univariate Analysis			Multivariate Analysis		
			HR	95%CI	*p* Value	HR	95%CI	*p* Value
DFS	Preop. CA 19-9 (log scale)		1.00	0.78–1.26	0.989	1.02	0.81–1.30	0.845
	TNM stage	III	4.94	1.59–7.35	<0.001	4.15	2.15–8.00	<0.001
		II	3.42	2.74–8.2	0.002	3.09	1.32–7.23	0.009
		I	(reference)			(reference)		
	R1 resection		1.24	0.74–2.10	0.420	1.35	0.67–2.68	0.396
	Adjuvant chemotherapy		1.40	0.76–2.59	0.264	0.91	0.46–1.82	0.792
	Sinistral portal hypertension		1.21	0.71–2.09	0.479	1.21	0.69–2.14	0.509
OS	Preop. CA 19-9 (log scale)		1.15	0.85–1.55	0.371	1.20	0.89–1.62	0.233
	TNM stage	III	4.45	2.42–8.19	<0.001	3.32	1.53–7.21	0.002
		II	3.47	1.52–7.91	0.003	2.92	0.69–7.05	0.180
		I	(reference)			(reference)		
	R1 resection		1.27	0.66–2.43	0.479	1.18	0.63–2.21	0.609
	Adjuvant chemotherapy		1.31	0.61–2.82	0.495	0.88	0.38–2.03	0.756
	Sinistral portal hypertension		1.32	0.76–2.29	0.328	1.03	0.62–1.73	0.902

HR, hazard ratio; CI, confidence interval; DFS, disease-free survival; OS, overall survival.

## Data Availability

The data presented in this study is available upon request from the corresponding author.

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
