# Peer review of "Should the Splenic Vein Be Preserved—Fate of Sinistral Portal Hypertension after Pancreatoduodenectomy with Vascular Re-Section for Pancreatic Cancer"

_cancers, 2022, doi:10.3390/cancers14194853_

Round 1
Reviewer 1 Report
Dr. Sung et al analyze the impact of splenic vein ligation on left-sided portal hypertension. Splenic vein ligation is sometimes necessary during pancreatic resections with vascular involvement, widely accepted by surgical community and supported by consistent literature. Interestingly, this article focuses on variceal formation and evolution during post-operative follow-up, analyzing its impact on surgical outcomes (short- and long-term) and survival, demonstrating no significant differences in SV ligation vs no ligation.
However some important issues should be addressed :
- The manuscript is strongly focused on technical surgical aspects rather than oncological topics; furthermore, oncological outcome is a minor part of analysys with no differences between the groups.
- Authors stated that SV ligation can lead to left-sided portal hypertension: How many patients developed SPH? All the SV ligation group? Please clarify
- Considering SMV/PV resection and incidence of strictures in the sample, information about vein reconstruction should be added and analyzed (type I,II,III,IV);
- Authors analyzed esopageal and gastric varices as main vessels to determine the variceal score; which is the status of left gastric vein in SV ligation group? This vein could be tributary to both PV and SV and it's preservation or reconstruction could be crucial, if no other gastric veins are preserved during pancreatoduodenectomy with vascular resection; data about left gastric vein should be reported and analyzed;
- Were there anatomic variants of regional vascularization? Author should clarify;
- Were there associated resections within the groups (i.e colonic or gastric associated resections)?
- Authors reported 2 cases of variceal bleeding: which varices bleeded? how were bleedings treated?
Conclusions: this article focuses on interesting surgical and post-surgical aspects of pancreatoduodenectomy with vascular resections, analyzing evolution of SV ligation that is a well-accepted condition. The manuscript could add some evidence about this particular surgical condition, but it should be strongly improved.
Author Response
Dear Editor,
Thank you so much for being so interested in our article. We carefully read the reviewers’ comments and appreciate their insightful comments. Our responses to the reviewers’ comments are as follows. For smoother explanations, the order of the reviewers’ comments was rearranged.
Reviewer 1
Comments and Suggestions for Authors
Dr. Sung et al analyze the impact of splenic vein ligation on left-sided portal hypertension. Splenic vein ligation is sometimes necessary during pancreatic resections with vascular involvement, widely accepted by surgical community and supported by consistent literature. Interestingly, this article focuses on variceal formation and evolution during post-operative follow-up, analyzing its impact on surgical outcomes (short- and long-term) and survival, demonstrating no significant differences in SV ligation vs no ligation.
However some important issues should be addressed :
- The manuscript is strongly focused on technical surgical aspects rather than oncological topics; furthermore, oncological outcome is a minor part of analysys with no differences between the groups.
Authors stated that SV ligation can lead to left-sided portal hypertension: How many patients developed SPH? All the SV ligation group? Please clarify
Response>
As described in the Materials and methods section, the variceal formation was regarded as present when the sum of variceal scores was higher than 4 on the postoperative six-month CT. According to the definition, 22 (23.4%) patients showed variceal formation after the surgery. In addition, the patients in the SV ligation group showed more variceal formation than those in the SV saving group (28.1% vs. 5.6%, p=0.028). We add more descriptions in the Results section including Tables.
- Considering SMV/PV resection and incidence of strictures in the sample, information about vein reconstruction should be added and analyzed (type I,II,III,IV);
Response>
As your comments, we reviewed the vein reconstruction type. In the tangential resection group, all patients underwent reconstruction using transverse suture without patch closure. In addition, 2 (3.1%) patients underwent vein reconstruction using an autologous grafted vein and the other 63 (96.9%) patients underwent end-to-end vein anastomosis. Although we tried to analyze according to vein reconstruction type, we could not conclude statistical results due to lack of sample size. We add these results in the Results section.
- Authors analyzed esopageal and gastric varices as main vessels to determine the variceal score; which is the status of left gastric vein in SV ligation group? This vein could be tributary to both PV and SV and it's preservation or reconstruction could be crucial, if no other gastric veins are preserved during pancreatoduodenectomy with vascular resection; data about left gastric vein should be reported and analyzed;
Response>
As you know, during PD, coronary vein was preserved normally. As your comments, we reviewed the operative records and there was no patient who experienced ligation of coronary vein. However, this study was designed retrospectively and we were not sure. Unfortunately, it was hard to distinguish the status of coronary vein after the PD due to postoperative change around the operative field. It was one of the limitation of this study and was stated in Discussion section.
- Were there anatomic variants of regional vascularization? Author should clarify;
Response>
As you know, venous variation around PV-SMV confluence was classified using IMV insertion type. Reviewing the arterial variation of the enrolled patients, 14 (14.9%) patients were revealed that their right hepatic artery arising from SMA and this rate was within the general range that is known to 10-15%. We add this result in the Results section.
- Were there associated resections within the groups (i.e colonic or gastric associated resections)?
Response>
Reviewing the operative notes, there was no case with combined organ resection. It was supposed that this study was retrospective and that the patients who underwent surgery would be well selected. In other words, perhaps the patients who were required combined organ resection were not considered candidates. We stated this in the Results section.
- Authors reported 2 cases of variceal bleeding: which varices bleeded? how were bleedings treated?
Response>
Fortunately, the patients’ bleeding was controlled using VDA. We stated this in the Results section.

Reviewer 2 Report
I have with great interest read this original article regarding the role ligation of the splenic vein has for SPH during pancraticoduodenectomy (PD) with vascular resections. This is an interesting topic with the potential to improve the surgery during PD with vascular resections. However, several questions arose when I read the manuscript.
1- Since the term sinistral portal hypertension (SPH) is included in both the title and article a clear definitions is needed, not just a reference.
2- I understand why you have excluded pts with no 6-months postoperative CT. However, this generates several questions which needs to be addressed. What was the routine, CT 6 months for everyone with PD + vascular resection or on demand? Did some pts succumb due to SPH before 6 months and therefore do not have a 6 months CT? Information about the these excluded pts is lacking
3- How many pts already had varices at the preoperative CT, this is important since varices can develop due to the tumors vascular involvement. In some cases they recede but not all, more information regarding the preoperative CT is needed
4- As I understand it the postop stricture was independently associated with the development of varices and was therefore excluded. However information regarding these 26 pts must be included. Did they have a ligation of the SV? Could it actually be that they got a stricture due to anastomotic tension with an intact SV?
5- Information from the pathology report raises several questions. 38% had a T0 or T1 tumor, did they really need a vascular resection? Only 19 % R1-resections. Since the modern definition of R1 is <1 mm and the thickness of the wall of the portal/SMV vein is <1 mm a majority of the pts that needs a vein resection is expected to have R1-resection. Did you use the old definition for radicality? I also find the amount of N2-tumours only 15 % for pts needing a vein resection? Here I want to address a typo it should be 9.3% N2 in SV ligation group (3/32) not 30.8%. Unfortunately I must draw the conclusion that either the pathology is suboptimal or these patients did not need a vascular resection. This needs a thorough rework during the discussion.
6- Since English is not my native language I have not done a complete review of the language, but I find it easy to follow and have no problems with the language.
Although there are several flaws in this study I still feel that the material with a major revision can contribute to the field and become worthy of publication.
Regards
Author Response
Dear Editor,
Thank you so much for being so interested in our article. We carefully read the reviewers’ comments and appreciate their insightful comments. Our responses to the reviewers’ comments are as follows. For smoother explanations, the order of the reviewers’ comments was rearranged.
Reviewer 2
Comments and Suggestions for Authors
I have with great interest read this original article regarding the role ligation of the splenic vein has for SPH during pancraticoduodenectomy (PD) with vascular resections. This is an interesting topic with the potential to improve the surgery during PD with vascular resections. However, several questions arose when I read the manuscript.
- Since the term sinistral portal hypertension (SPH) is included in both the title and article a clear definitions is needed, not just a reference.
Response>
In this study, SPH was defined as the occurrence of left-sided variceal formation and the formation was regarded as present when the sum of variceal scores was higher than 4 on the postoperative six-month CT. We stated this in Materials and methods section.
- I understand why you have excluded pts with no 6-months postoperative CT. However, this generates several questions which needs to be addressed. What was the routine, CT 6 months for everyone with PD + vascular resection or on demand? Did some pts succumb due to SPH before 6 months and therefore do not have a 6 months CT? Information about the these excluded pts is lacking
Response>
As your comment, we reviewed the medical records of the patients who were excluded and found 9 patients. 7 patients revealed early recurrence and did not want additional therapy; the other 2 patients were lost to follow-up. However, there was no variceal bleeding event during their follow-up periods. We added the supplementary explanation in the Materials and methods section for a clearer study design.
- How many pts already had varices at the preoperative CT, this is important since varices can develop due to the tumors vascular involvement. In some cases they recede but not all, more information regarding the preoperative CT is needed
Response>
As described in the Materials and methods section, variceal formation was regarded as present when the sum of variceal scores was higher than 4. Although a few patients’ scores were not zero on preoperative CT, their scores did not exceed 4. In other words, no patients showed variceal formation before the surgery. We added the supplementary explanation in the Materials and methods section for more clear study design.
- As I understand it the postop stricture was independently associated with the development of varices and was therefore excluded. However information regarding these 26 pts must be included. Did they have a ligation of the SV? Could it actually be that they got a stricture due to anastomotic tension with an intact SV?
Response>
As your comments, we reviewed the 26 patients. Fifteen patients were in SV saving group and the other 11 patients were in SV ligation group. We added this data in the Results section. In addition, vascular anastomotic tension might induce stricture. However, before the vascular reconstruction, PV-SMV confluence area was dissected as much as possible to achieve tension-free status. No data could prove the hypothesis due to its retrospective design. It is a limitation of this study.
- Information from the pathology report raises several questions. 38% had a T0 or T1 tumor, did they really need a vascular resection? Only 19 % R1-resections. Since the modern definition of R1 is <1 mm and the thickness of the wall of the portal/SMV vein is <1 mm a majority of the pts that needs a vein resection is expected to have R1-resection. Did you use the old definition for radicality? I also find the amount of N2-tumours only 15 % for pts needing a vein resection? Here I want to address a typo it should be 9.3% N2 in SV ligation group (3/32) not 30.8%. Unfortunately I must draw the conclusion that either the pathology is suboptimal or these patients did not need a vascular resection. This needs a thorough rework during the discussion.
Response>
First, thank you for your comment and we revised the table data correctly.
About half of the enrolled patients underwent neoadjuvant CTx. in this study. As a result, around the tumor area was shown fibrotic change in the operative field during the surgery and we could not discriminate between fibrotic adhesion and tumor invasion of vascular structure at the time of the surgery. Although we carefully dissected vascular structure, sometimes the pancreas could not separate from the vascular structure. In that situation, we decided to perform vascular resection because we were unsure of its character.
As you know, recent data showed that resection margin between 0 and 1 mm was a poorer prognostic factor than resection margin beyond 1 mm. However, it is still a controversial issue and there is no clear consensus regarding vascular resection. In addition, the concept of N2 for pancreatic cancer was introduced in AJCC 8th edition and there were only N0 and N1 in the previous editions. Therefore, we did not consider the nodal factor at the time of the surgery. Above all, this study was intensely focused on technical surgical aspects rather than oncological topics. We added the comments in the Discussion section.
6- Since English is not my native language I have not done a complete review of the language, but I find it easy to follow and have no problems with the language.
Response>
Actually, we had a native speaker grammatically edit this manuscript. However, additional corrections will be performed if the manuscript needs further proofreading.
Reviewer 3 Report
The paper entitled “Should the splenic vein be preserved?: Fate of sinistral portal hypertension after pancreatoduodenectomy with vascular resection for pancreatic cancer” presents a very interesting study on sinistral portal hypertension after pancreatoduodenectomy. Nevertheless, it requires several additions and clarifications.
It must be specified whether the Whipple procedure or Traverso (the pylorus-preserving pancreatoduodenectomy) was performed.
It should also be specified how many of the 94 patients had their superior mesenteric vein resected and how many underwent portal vein resection. Vessel resection indicates that surgical exploration confirmed vascular infiltration. Please, specify how many patients had resectable tumors and how many suffered from borderline resectable disease (according to NCCN 2017, 2022 or according to number 5 citation NCCN 2021).
Please, specify which TNM staging (which year?) was used (in the Material and Methods Section) and provide relevant literature references. Table 1 shows T stage 0 was seen in 6 (6.4%) of the patients. Why was vascular resection performed in these patients and in TNM 0 (Tis N0 M0)? [The 8th edition of the UICC TNM Classification]
In 17 patients , R1 resection was achieved despite vascular resection. Which margins were positive (R1)?
The authors report 94 consecutive patients underwent pancreatoduodenectomy with vascular resection for pancreatic cancer. Please, specify the postoperative mortality rate and how many patients (n) were analyzed at 6, 12 and 24 months postoperatively.
Please, specify the year the Clavien-Dindo and POPF classifications were devised and include this information in the Material and Methods Section.
Overall survival depends on the development of postoperative complications and this factor was not analyzed. The fact should be acknowledged in the limitations of the study.
The authors should be congratulated on excellent outcomes – only two fistulas occurred (B and C) (2.1%) !
The Introduction Section contains a sentence : “During the inevitable vascular resection, SV ligation can induce sinistral portal hypertension (SPH), or left-sided portal hypertension.” I suggest rephrasing the sentence to “During the inevitable vascular resection, SV ligation can induce sinistral portal hypertension (SPH) otherwise referred to as left-sided portal hypertension.”
Author Response
Dear Editor,
Thank you so much for being so interested in our article. We carefully read the reviewers’ comments and appreciate their insightful comments. Our responses to the reviewers’ comments are as follows. For smoother explanations, the order of the reviewers’ comments was rearranged.
Reviewer 3
Comments and Suggestions for Authors
The paper entitled “Should the splenic vein be preserved?: Fate of sinistral portal hypertension after pancreatoduodenectomy with vascular resection for pancreatic cancer” presents a very interesting study on sinistral portal hypertension after pancreatoduodenectomy. Nevertheless, it requires several additions and clarifications.
It must be specified whether the Whipple procedure or Traverso (the pylorus-preserving pancreatoduodenectomy) was performed.
Response>
All enrolled patients underwent PPPD and we rewrote the Materials and methods section for more clear study design.
It should also be specified how many of the 94 patients had their superior mesenteric vein resected and how many underwent portal vein resection. Vessel resection indicates that surgical exploration confirmed vascular infiltration. Please, specify how many patients had resectable tumors and how many suffered from borderline resectable disease (according to NCCN 2017, 2022 or according to number 5 citation NCCN 2021).
Please, specify which TNM staging (which year?) was used (in the Material and Methods Section) and provide relevant literature references. Table 1 shows T stage 0 was seen in 6 (6.4%) of the patients. Why was vascular resection performed in these patients and in TNM 0 (Tis N0 M0)? [The 8th edition of the UICC TNM Classification]
Response>
We referred to AJCC 8th edition and added the manuscript including reference.
About half of the enrolled patients underwent neoadjuvant CTx. in this study. As a result, around the tumor area was shown fibrotic change in the operative field during the surgery and we could not discriminate between fibrotic adhesion and tumor invasion of the vascular structure at the time of the surgery. Although we carefully dissected vascular structure, sometimes the pancreas could not separate from the vascular structure. In that situation, we decided to perform vascular resection because we were unsure of its character. We added the comments in the Discussion section.
In 17 patients , R1 resection was achieved despite vascular resection. Which margins were positive (R1)?
Response>
In detail, R1 patients were 11 of the reasons for circumferential margin positive, followed by p-duct positive with 5 patients, and vascular resection margin with 1 patient. Although the frozen reports of all p-duct positive patients were negative during the surgery, the reports were changed to margin positive in the permanent section reports.
The authors report 94 consecutive patients underwent pancreatoduodenectomy with vascular resection for pancreatic cancer. Please, specify the postoperative mortality rate and how many patients (n) were analyzed at 6, 12 and 24 months postoperatively.
Response>
Although 4 patients underwent ICU care due to septic events (related to POPF or bleeding), there was no mortality case in this study, fortunately. We are guessing it reflected selection bias for the surgery decision. Considering the patient’s morbidity, we selected operable and resectable patients. We added these points in the Results and Discussion section.
Please, specify the year the Clavien-Dindo and POPF classifications were devised and include this information in the Material and Methods Section.
Response>
We referred to the 2016 updated POPF definition and added the manuscript including related references.
Overall survival depends on the development of postoperative complications and this factor was not analyzed. The fact should be acknowledged in the limitations of the study.
Response>
As your comment, we added the detailed limitation in the Discussion section.
The authors should be congratulated on excellent outcomes – only two fistulas occurred (B and C) (2.1%) !
Response>
As we stated previously, this study might have selection bias for the surgery decision. For this reason, favorable outcomes were reported. In addition, all enrolled patients were pancreatic cancer patients. As you know, dilated p-duct and hard pancreas were good prognostic factors of POPF. Most of the enrolled patients showed these conditions and we guessed that although pancreatic resection was too difficult, the reconstruction phase was done smoothly, relatively.
The Introduction Section contains a sentence : “During the inevitable vascular resection, SV ligation can induce sinistral portal hypertension (SPH), or left-sided portal hypertension.” I suggest rephrasing the sentence to “During the inevitable vascular resection, SV ligation can induce sinistral portal hypertension (SPH) otherwise referred to as left-sided portal hypertension.”
Response>
We exchanged the sentence on your advice.

Round 2
Reviewer 1 Report
The manuscript has been deeply improved, addressing all issues previously reported. Oncological aspects have been expanded. It's suitable for publication.
Reviewer 2 Report
I have with great interest read revised version of the original article regarding the role ligation of the splenic vein has for SPH during pancraticoduodenectomy (PD) with vascular resections. This is an interesting topic with the potential to improve the surgery during PD with vascular resections. In my review I suggested revisions which have now been done. I think that my comments have been adequately met and answered and that the manuscript is ready for publication in the present form. As I wrote before I must point out that English is not my native language and I have not done a complete review of the language, but I find it easy to follow and have no problems with the language.
Kind regards